# Pediatric Perspectives and Tools for Attorneys Representing Immigrant Children: Conducting Trauma-Informed Interviews of Children from Mexico and Central America

**Ryan B. Matlow** [1,*]**, Alan Shapiro** [2] **and N. Ewen Wang** [3]

[1] Department of Psychiatry and Behavioral Sciences, Stanford University School of Medicine, Stanford, CA 94305, USA
[2] Department of Pediatrics, Montefiore Medical Center, Terra Firma National, New York, NY 10459, USA
[3] Department of Emergency Medicine, Stanford University School of Medicine, Stanford, CA 94305, USA
* Correspondence: rmatlow@stanford.edu

**Abstract:** Pediatric health and mental health professionals with expertise in the physical and emotional needs of immigrant children seeking humanitarian protection are trained to understand and address the sometimes deeply traumatic nature of their experience. This expertise plays an important role in collaborating with immigration attorneys to provide compassionate, trauma-informed representation that centers on children's best interests. In medicine, we say that "children are not small adults," such that meeting a child's needs requires consideration of their developmental stage and the unique impacts of child trauma exposure. This also holds true for legal professionals dedicated to protecting the rights of children in migration. This article aims to (1) review the principles of trauma-informed care in the context of child development, (2) understand the traumatic nature of the migration paradigm for children from Mexico and Central America seeking safety and protection, and (3) suggest ways that healthcare, mental health and legal professionals can inform one another's efforts to optimize the wellbeing of children and improve legal outcomes. The application of this knowledge in practice can advance legal goals, reduce risk for child re-traumatization during interviews, and reinforce child strengths while also reducing vicarious trauma and burnout for legal professionals.

**Keywords:** children; trauma; interviewing; migration; asylum

## 1. Introduction

The purpose of this article is to bring awareness to the plight of immigrant children from Mexico and Central America who seek safety in the United States, acknowledge the compounded trauma that many experience, explain the physiologic and psychologic effects of trauma, and provide insight into a trauma-informed approach for engaging and conducting legal interviews with these children. The authors of this paper believe this knowledge and practice approach can lead to improved communication and legal—as well as health and psychological—outcomes for children and families seeking durable legal status[1] in the US. These approaches also address the best interests of the child, prevent

---

[1] Children and families seeking refuge and asylum in the United States pursue an official, authorized legal status as a means of ensuring their safety and protection. Here, we use the term "durable legal status" to convey the need and benefit of securing long-term protections, in recognition of the fact that many legal status classifications are temporary or short-term in nature (e.g., temporary protected status, protection against deportation during an appeal), thereby failing to protect the best interests and well-being of children. There are multiple routes or processes for securing durable legal status (e.g., before an immigration judge in an asylum hearing, or before a USCIS officer in an affirmative process); our article aims to broadly cover the principles and practices of trauma-informed interviewing that are generally relevant to legal professionals interacting with immigrant children, however, we acknowledge that strategies may vary depending on children's specific circumstances and context within the U.S. immigration system.

re-traumatization, and facilitate a positive trajectory that can allow children and families to reach their full potential. While we focus primarily on the experience of immigrant children from Mexico and Central America seeking safety in the US (as they make up the majority of recent arrivals), the knowledge, principles, and practices discussed can and should also apply to unaccompanied children and those in families who come from other parts of the world seeking humanitarian protection in the US. These principles apply to children worldwide who flee their communities of origin and seek safety in other countries.

Legal and advocacy interviews with immigrant children often serve the essential and critical function of helping them secure legal status, which in itself improves myriad outcomes for the child, as immigration status is a recognized social determinant of health (Castañeda et al. 2015). However, these interviews are not conducted without risk, as the content and focus are inherently sensitive and can elicit psychological and physical distress. Children's negative experiences in interviews can deter them from continued pursuit of their legal interests, and can also potentially cause lasting psychological harm, with risk for reinforcing or exacerbating prior negative experiences many children have had with government officials and representatives. Recently arrived immigrant children (or, those who are more generally lacking durable legal status that provides long-term protection against removal) are inherently in a critical and sensitive circumstance of uncertainty, risk, and adjustment; thus, professionals engaging with these children have an important opportunity and responsibility to offer a supportive, welcoming, child-friendly and culturally sensitive experience that protects children's best interests and reinforces their existing strengths and resiliency. These are the goals of a trauma-informed interview.

Legal professionals may interact with immigrant children in a variety of contexts. Once released into U.S. communities, many children will access community-based legal services as they seek support with their legal claims. Other professionals and advocates will encounter children while they are in government custody, to offer screening, general information and advice (e.g., "know your rights"), as well as crisis or acute legal representation. Still others may engage with children in international contexts as children approach U.S. ports of entry. We have worked with legal professionals in each of these contexts, and we recognize that each situation can present its own challenges and constraints for the interview and advocacy process. These can include: restrictions on meeting times for attorneys while children are in government facilities, inability to modify the meeting space while children are in international shelter settings or in government custody, or difficulty following up with children who are dispersed in the community. While the context of engagement may significantly constrain or expand practice approaches (including the opportunities to advance a trauma-informed approach), there are universal principles and guidelines for trauma-informed practice that should be maximized within the constraints of the environment and context. Engaging in this trauma-informed approach begins with finding out about the child's migration experiences, understanding the child's developmental stage, and the potential complex impacts of multiple and severe trauma exposure. This knowledge serves as the foundation for the application of specific interview practice approaches. The current article aims to provide an introduction to the core knowledge base critical for working with immigrant children seeking protection in the US, provide an overview of practice recommendations for trauma-informed interviewing, and make a call for interdisciplinary collaboration and consultation in advocacy and support efforts with this population. With this background and spirit, we hope to advance a welcoming, strengths-based approach, as well as a system of reception that protects the universal rights and best interests of immigrant children.

## 2. Immigrant Children Seeking Protection in the United States

The past decade has seen major social upheaval with unprecedented levels of population displacement and mass migration. War, abject poverty, social inequities, lack of government protection, and climate crises are among the driving factors causing millions to flee for safety and fight for survival. According to the UN High Commissioner for Refugees,

there were 89 million people forcibly displaced globally by the end of 2021 and of these 36.5 million (41%) were children below the age of 18 years. By mid-2022, there were 32 million refugees among the 103 million forcibly displaced (UNHCR Refugee Data Finder 2022). The United States has been experiencing the effects of mass migration as hundreds of thousands, predominantly from northern Central America and Mexico make their way to the U.S. southern border. With almost 900,000 individuals forcibly displaced from Northern Central America (UNHCR Refugee Statistics 2022), scores of unaccompanied children and families (predominantly women and children) turn to the U.S. southern border as a destination for refuge and protection. Since 2012, millions of unaccompanied children and members of family units have sought humanitarian protection at the southwest U.S. border (United States Customs and Border Patrol 2022; see Figures 1 and 2).

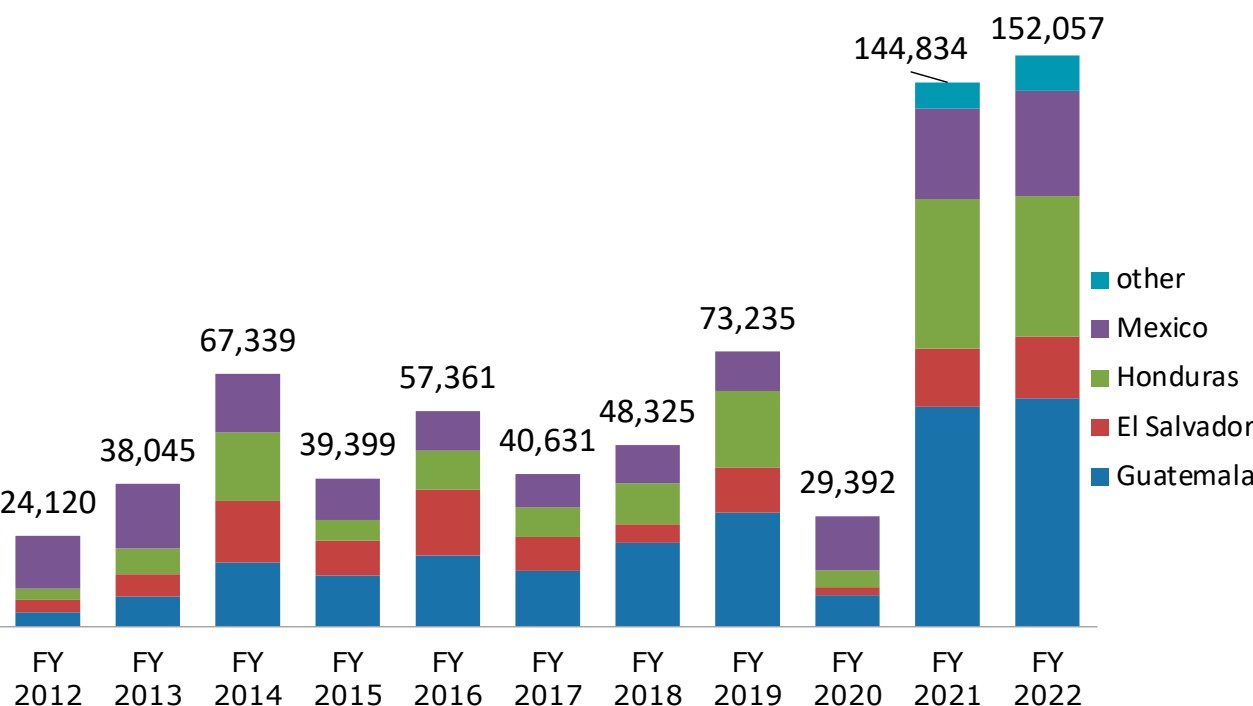

**Figure 1.** Number of unaccompanied children apprehended at the Southwest Border from 2012–22. (Source: U.S. Customs and Border Protection: Southwest Land Border Encounters; https://www.cbp. gov/newsroom/stats/southwest-land-border-encounters, accessed on 30 December 2022).

Major factors driving mass migration from Northern Central America ('push' factors) include pervasive community violence perpetrated by gangs and drug smugglers, domestic violence with little to no protection for children and women, and abject poverty made worse during the COVID pandemic and natural disasters. Sociologists, political scientists, and anthropologists blame ineffectual governmental infrastructures, weak to non-existent judicial systems, economic collapse, and intractable corruption that provide scant protection for the most vulnerable members of these countries. Similar factors including political instability and economic collapse have driven a rapidly growing displaced community from additional countries, such as Venezuela. Important factors influencing migration to the US (pull factors) include reunification with family members (as the clear majority of unaccompanied children have family already in the US) (Matlow et al. 2021), educational and economic opportunities and a search for safety and justice. Although the rise in the number

of unaccompanied children and families began a decade ago, a systematic implementation of a child-centered, trauma-sensitive approach has yet to be implemented despite calls from experts and advocates (Linton et al. 2017). Border apprehensions, custody placements, and administrative processing continue to treat unaccompanied children and families seeking U.S. protection as presumptively unfit and unreliable, subjecting them to the adversarial and restrictive processes that are characteristic of criminal prosecution. Prolonged detention in unsafe facilities, family separation, and poor access to healthcare, mental health services and legal counsel are among the significant problems in the current system.

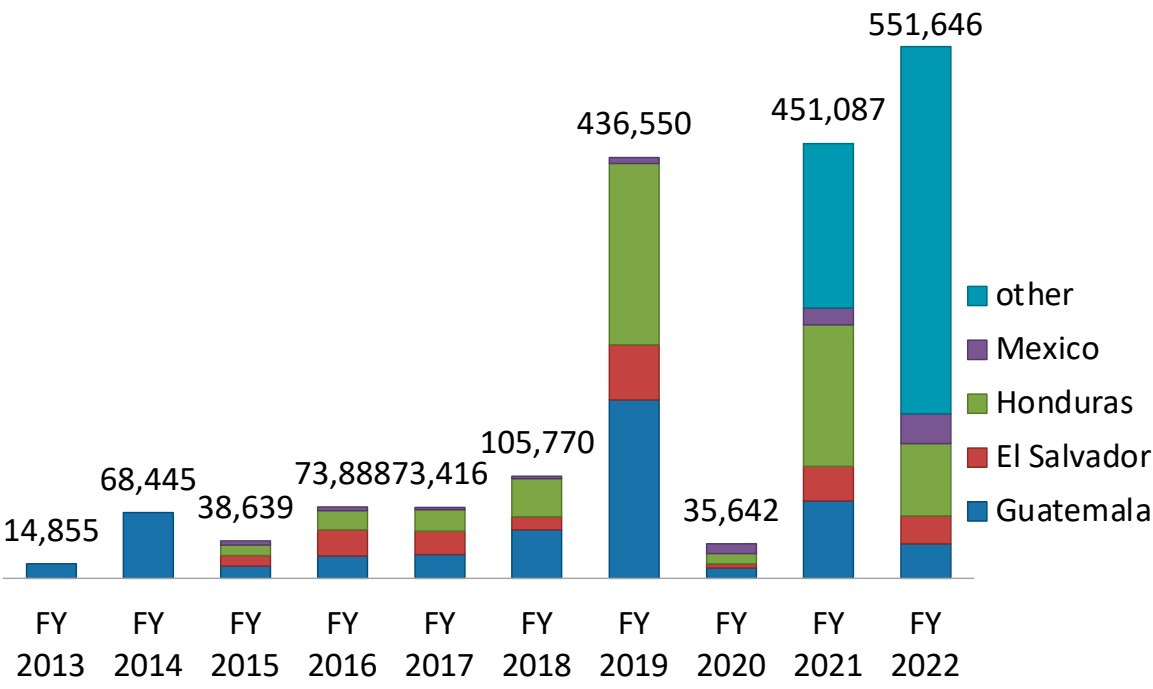

**Figure 2.** Number of individuals apprehended in family units at the Southwest Border from 2013–21. (Source: U.S. Customs and Border Protection: Southwest Land Border Encounters; https://www.cbp.gov/newsroom/stats/southwest-land-border-encounters, accessed on 30 December 2022).

### 2.1. The Migration Paradigm: Stages of Migration and Associated Experiences of Adversity and Trauma

Professionals of any discipline that work with immigrant children should adopt a strengths-based approach to demonstrate understanding of children's resilience and avoid characterizing them as victims. With that said, it is also important to recognize that many children fleeing to the US have been witnesses or survivors of violence and other forms of trauma. A comprehensive approach to understanding traumatic histories of unaccompanied children and those traveling with caregivers, is to break down their experiences into four unique stages of migration: pre-migration (country of origin), migration (journey to the U.S. border), apprehension and detention (crossing into the US and being taken into custody by immigration officers) and post-detention (release and integration into U.S. communities while in immigration court proceedings).

Once released from U.S. immigration custody, children and their caregivers begin the complex process of acculturation and integration into their new communities. They will likely interact with a wide range of professionals that include but are not limited to immigration attorneys, healthcare workers, social service specialists, and educators. Understanding this migration framework can guide the exploration of a child's experiences during trauma sensitive interviewing (see below). An appreciation of a child's resilience and trauma can go a long way in building trust and strengthening relationships. It can also provide a basis for comprehending (and empathizing with) behaviors that may at times seem to undermine their clients' best interests. The following sections will focus on some of the ways that children experience trauma through the four stages of migration, with case examples to illustrate.

### 2.1.1. Pre-Migration

Rigoberto's story (Box 1) highlights important antecedents that cause children to flee—in this case, the persecution and exploitation of impoverished youths from ethnic minority groups. The story of Rigoberto underscores the multiple levels of hardship and trauma that many youth face in their home countries. Children fleeing from Central America and Mexico cite community and domestic violence and lack of government protection as the principal factors leading them to flee their country (Keller et al. 2017; Doctors Without Borders 2020; Sidamon-Eristoff et al. 2022). Gangs and other criminal elements control many aspects of society and prey on the vulnerable, specifically unprotected children, girls and women, and ethnic, racial and sexual minorities. Transnational gangs are widespread through northern Central America and their structure is one of decentralization, bringing violence to all areas of these countries (Paarlberg 2022). Additionally, poverty, war, natural disasters and persecution lead to family fragmentation as adult members of the family are forced to seek safety and/or economic opportunities elsewhere in their own country and abroad. Loss of a parent(s) places children at heightened risk for community violence as well as domestic violence such as physical, sexual, and verbal abuse, and neglect. This reality is underscored by studies conducted by human rights groups, which have identified threats by gangs, drug cartels, targeted violence, child abuse in the home, as well as general domestic and community violence as primary reasons why children fled their countries (Women's Refugee Commission 2012; UNHCR 2014).

**Box 1.** Case example.

> Rigoberto is a 17-year-old youth who was raised in an Indigenous Mam community in Guatemala. His father was left with serious disabilities after an accident at a construction site. Rigoberto was forced to leave school at a young age to work and help feed his family. He recounted how he was hired for a job on a coffee plantation but soon realized that this "job" was in reality forced labor. He and other indigenous young men and women were unable to leave. Their every move was watched by security guards. Attempts at escape were met with harsh punishment and he was made to watch a fellow laborer's fingers cut off after that worker unsuccessfully tried to escape. Rigoberto worked for 6 months at the plantation, was never paid and described his experience as slavery. One night, after the security guards had been drinking heavily, he escaped and made a harrowing journey trekking through jungle on his way back to his home village. (He is still covered in scars from injuries he sustained from that escape.) Once back in his village, he learned that the landowners of the plantation were searching to kill him. With money given to him by his family and with little preparation, he fled Guatemala in order to escape death.

Over the past few years, these trends have remained unchanged as northern Central American countries (especially Honduras) and Mexico continue to experience rising violence (International Rescue Committee 2022), which has been exacerbated by the COVID-19 pandemic (Center for Strategic and International Studies 2022) and climate disasters (e.g., sequential hurricanes Eta and Iota devastated northern Central America in November 2020, and caused over a billion of dollars of loss; Mughal and S&P Global Market Intelligence 2022; U.S Agency for International Development 2022). Local governments fail to provide

adequate protection as they continue to be saddled by entrenched corruption (International Rescue Committee 2022). These deteriorating conditions are reflected in the steady increases in the numbers of unaccompanied children and families that are seeking safe haven in the US.

In other parts of the world, 2021 saw tremendous upheaval as political instability and oppression, economic deterioration, and war drive new waves of displacement and migration such as in Afghanistan (UNHCR 2021 Global Report, Afghanistan Situation 2021), the Russian-Ukrainian war (Yayboke 2022) and Venezuela (UNHCR Venezuela Situation 2022). Many of the children and families displaced by these global events arrive at the U.S. Southern border (along with those from Mexico and Central America) seeking safety, protection, and refuge.

Fiscal year 2022 saw the greatest number of unaccompanied children (152,057) and members of families (551,646) seeking protection at our border in U.S. history (United States Customs and Border Patrol 2022). This followed on the heels of FY2021, the year that previously held the highest record of apprehensions of unaccompanied children and members of family units.

### 2.1.2. Migration and Border Crossing

Trauma associated with these perilous journeys to the U.S. border is often multifaceted (Box 2). The decision of a child to flee one's home, with or without parental/guardian consent, may be followed by a sense of loss of family, friends and community. The arduous nature of the journey gives children little time to process that loss. Like unaccompanied children the world over, those from Central America and Mexico are highly vulnerable along their travel routes and are at risk—and often specifically targeted—for robbery, kidnapping, exploitation, as well as, physical and sexual victimization. Those who travel with guides (*coyotes*) may have some protective advantage but may also be victimized by them (Gilardi 2020). In order to evade Mexican authorities, migrant children and families resort to dangerous forms of transportation such as riding atop freight trains (*la bestia*) or packed in the back of trucks such as the child in the case example (Box 2). Many must walk long distances enduring the elements and are at risk for significant exposure and injury. Most migrant children and families will suffer from hunger and sleep deprivation, and often have little access to medical care on their journey. These dangers listed above are compounded and multiplied for those traveling from other continents—often taking life threatening routes through the notorious lawless and perilous Darién Gap in Panama (Roy 2022); through which greater than 70,000 migrants passed in 2021 (Suarez 2021). Adolescent girls, adult women, sexual minorities, and racial and ethnic minorities are among the most vulnerable migrants and face serious risk of sexual victimization and exploitation. Finally, anti-migrant, anti-immigrant U.S. policy that disregards the unique needs of children, pregnant women, families and other vulnerable populations have made seeking safe haven increasingly more difficult for years. Most recently, the Migrant Protection Protocols and Title 42, a pandemic-related public health declaration has left hundreds of thousands of migrants—including children and families—stranded in dangerous border towns in Mexico. These policies place vulnerable migrants at heightened risk for victimization, including kidnapping, robbery, assaults, rape, malnutrition and illness (Human Rights Watch 2021). It is important to note that while the majority of children presenting at the U.S. Southern border are from Mexico and Central America, an increasing number of children and migrants are coming from other regions of the world, including Haiti, Africa and more recently the Ukraine.

**Box 2.** Case example.

> Margarita is a 16-year-old girl who fled from El Salvador because gang members killed her boyfriend and then threatened to kill her. She fled El Salvador without telling her grandmother with whom she lived, and traveled to the U.S. border with a guide (coyote) and 10 other adults and children. She described the various forms of transportation on her arduous month-long journey that included local buses, walking miles and being packed into the back of a tractor trailer truck. She describes the truck as being so overcrowded with migrants that it was impossible to move. The trip lasted 12 h and with little air, stifling heat, and no food or water. She sobbed recalling that trip and how she was certain she and others would die. Luckily everyone in that truck survived and the group was taken to a *bodega*, "safe house," which she described as anything but safe. The men guarding the house got drunk at night and would choose female migrants with whom to have sex. Although she was not harmed physically or sexually, she says she carries the emotional scars of that experience.

With few ports of entry along the almost 2000 mile U.S.–Mexico border, many migrants have no choice but to enter the US by crossing the Rio Grande or desert. These can be dangerous, even fatal, crossings. The United Nation's International Organization for Migration reported that at least 650 migrants died trying to cross the U.S.–Mexico border in 2021; the highest number on record (International Organization for Migration 2021). This can be a terrifying leg of the journey for children.

While the vast majority of unaccompanied minors and families intend to present themselves to U.S. officials so they can ask for asylum once they set foot on US soil, some are forced to walk for hours if not days in dangerous terrain until they are found. This too places them at risk for illness, injury and exploitation such as kidnapping.

### 2.1.3. Apprehension and Detention

Myriad reports by human rights, immigrant and child advocates have detailed the inhumane and dangerous conditions of the U.S. immigration detention system for unaccompanied children and families (as well as single adults) (Box 3). These reports cite overcrowded, unsanitary, freezing cold facilities in which children are not provided nutritious foods and at times must sleep on cement floors where bright lights are never lowered (Linton et al. 2017; PBS News Hour 2019; O'Leary 2019). In addition to harsh conditions, the staff is made up of predominantly law enforcement officers, who have little to no experience in the care of children and families (or pregnant women) nor proper training in trauma-informed care. Professional healthcare organizations also have raised alarms on the lack of specialized pediatric healthcare providers and other child welfare professionals (Linton et al. 2017). Moreover, close to 4000 children were separated from their caregivers under the Trump-era Zero Tolerance policy (Ward 2021) although many experts suspect the number to be much higher. To date, there are over 1000 children who have yet to be reunited with their parents (Southern Poverty Law Center 2022).

**Box 3.** Case example.

> Genesis and her 6-year-old son, Francisco, fled their Garifuna community in Puerto Cortés, Honduras, after her boyfriend tried to kill her when he found out that she was HIV positive. After a three-week journey that included riding atop Mexican freight trains (*la bestia*), they crossed the Rio Grande on a raft and were apprehended by Border Patrol in the US. They were first taken to a large, windowless and cold warehouse-like facility (Customs and Border Protection processing center, also known as *la hielera*, or freezer). On the second day of detention, when Genesis returned from being interviewed by a customs officer, she discovered that Francisco was removed from the facility without her knowledge or permission, and sent to an Office of Refugee Resettlement shelter for unaccompanied minors in New York. Genesis was subsequently sent to an adult detention center in Washington State. She went weeks without receiving information about Francisco's placement or well-being. She states that the separation was the worst thing that ever happened to her and that feels she will never get over the pain of that traumatic experience. Although eventually reunified with Francisco, Genesis reports that her son has regressed so much that he now has to wear a diaper to bed and has stopped talking to almost anyone including herself. He experiences anxiety and panic when he encounters an official dressed in uniform and fears he will be separated again.

While some Trump-era policies, such as Zero Tolerance have been reversed and there has been some decrease in crowding, many of the concerning practices, and the general failures to invest in infrastructure that protects children's well-being, persist. Other US practices that may lead to re-traumatization include the transfer of unaccompanied children to influx facilities and emergency intake sites (EIS) when there is a surge at the border (Matlow et al. 2021). Children have reported being terrified by being transferred from one shelter to an influx site in the middle of the night with no warning. These congregate care facilities are highly variable in the level of protection and care provided to children, and they are largely exempt from child protection and service standards (based on state childcare licensing requirements) that apply to other shelters and custody settings (Matlow et al. 2021).

2.1.4. Post-Detention Community Release

Unaccompanied children and families released from immigration detention into the US enter a period of promise yet great transition as they wait for their immigration case to be adjudicated. During this period they face myriad health, social, educational and legal challenges (Boxes 4 and 5). Unaccompanied children often reunite with family they have not seen for years, if at all. They are thrust into new relationships and must begin the process of acculturation—learning a new language, new customs and forging a new identity. As many unaccompanied children and families resettle in under-resourced communities, they must negotiate this complex period of their life with little support or access to public benefits. Lack of health insurance creates enormous barriers to seeking the medical and mental health services they need. Currently, only six states (NY, MA, IL, WA, OR, CA) and the District of Columbia provide state-supported health insurance (S-Child Health Insurance Plan) for children under 18 years of age regardless of immigration status (National Immigration Law Center 2022). Restrictive U.S. policies delay work authorization thus leaving newly arrived immigrants with few opportunities to become self-reliant. This places recently arrived immigrants at great risk for food insecurity, housing instability, homelessness and social isolation, which can further exacerbate earlier trauma and lead to new psychological and medical problems.

**Box 4.** Case example.

Abdul is a 17-year-old youth from Ghana, who fled to the US after an attempt on his life by assailants hired by his father's first wife. As the first-born male to his father's second wife (Abdul's mother), he became first in line to inherit his father's land and cattle. Provoked by anger, jealousy and greed, his father's first wife arranged to have him killed. Abdul sustained life threatening injuries but survived and when he was released from a month-long hospitalization, he fled Ghana. He was given assistance by an uncle to escape to neighboring Togo and, from there, paid for his passage to Brazil. He traveled to the US where his maternal uncle lived with his wife and 2 children. After a 5-month journey through Central America and Mexico, in which he was robbed twice, once at knifepoint, he finally arrived to the U.S. border and sought asylum at a U.S. Customs Port of Entry. After spending 3 months in a Chicago shelter for unaccompanied minors, he was released to his uncle and his wife in the Bronx where he currently lives. He had not seen his uncle since he was a little boy and reports having difficulty building a relationship with him, and reported that his uncle's wife is particularly resentful of having to support him. Their apartment is crowded and they are having financial difficulty due to the COVID-19 pandemic. Although Abdul is enrolled in high school and is an excellent student, he finds studying increasingly more difficult as he has to work 6 h every day after school as a restaurant dishwasher. He is contemplating dropping out of school.

Abdul's case is a clear example of how legal status in itself is a social determinant of health; and being undocumented without access to healthcare or other social benefits can lead to deleterious outcomes (Castañeda et al. 2015). This basic tenet underscores the importance of persisting through the legal advocacy process and minimizing distress through this process. Durable legal status not only confers safety and security but also eligibility for public benefits that include housing support, Supplemental Nutrition Assistance Program (formerly known as food stamps), work authorization and health insurance.

These all-important benefits are part of the safety net that new immigrants need to land on solid ground, in order to increase their chances of success and promote integration as valuable members of our society.

Navigating new relationships between unaccompanied youth and their sponsors can be a complicated process. While the large majority of unaccompanied youth are unifying with parents or family members upon their arrival to the US, many have not seen each other in years, and some are getting to know each other for the first time. The reunion can lead to an initial "honeymoon" period marked by great happiness but then give way to guilt, anger and resentment on the parts of child, parent and other members of the child's new family (e.g., step-parent, new siblings). This is not surprising as factors related to both the sponsors' and children's migration-related traumas may raise issues of abandonment, loss, and lack of trust and protection. Unaccompanied youth may bristle at rules and restrictions imposed by the parent, or other adult sponsor, who did not have this role for much of their lives due to separation. This can lead to externalizing psychological symptoms (e.g., oppositional and defiant behaviors) and internalizing symptoms (e.g., depression and anxiety). In the absence of intervention, irreconcilable differences can lead to worsening psychological problems, learning problems, housing insecurity or homelessness; further complicating integration and acculturation. It is imperative that unaccompanied children and families be connected to a strong safety net of services in order to mitigate the considerable challenges outlined above (Greenberg et al. 2021).

The mere process of pursuing durable or permanent legal status can be an additional stressor. Children and their families live in a state of limbo, uncertainty, and chronic anxiety as they wait years for their case to come to trial or to reach final adjudication (i.e., whether before a judge or a USCIS immigration official). During this time, they understand the risk of being returned to their country of origin from which they escaped and the threats that may pose on their life. Further, the emotional stress of be interviewed by an immigration official or testifying before a judge cannot be overstated; knowing their testimony can make a difference in the final judgment of their case. Moreover, many times children will have to openly recount painful events to an unfamiliar official or judge that include painful and embarrassing details of abuse or torture. Anxiety around testifying also includes fear of reprisal (e.g., from transnational gang members or other family members in the US), fear of going to jail, and fears of deportation. Children whose initial requests for durable status are not approved (e.g., by an asylum officer) may experience notable distress and concern, but may still have opportunities to seek protection through further appeals or application processes. However, having to present their case again may add a new layer of stress especially as the child's immigration status remains in limbo and the requirement to recount their history once again may pose risk for re-traumatization.

**Box 5.** Common Post-release stressors.

- *Reunification and integration with family/sponsor*
- *Family/household conflict after "honeymoon"*
- *Exacerbated social stressors in sponsoring household (e.g., financial, housing, food)*
- *Carried over trauma—re-victimization*
- *Legal system—fear of deportation, stress in providing testimony*
- *Acculturation—identity shift, language barriers*
- *School system—unable to navigate*
- *Isolation—lack of community*
- *Discrimination, lack of sense of belonging*
- *Survivor's guilt—"carriers of hope"*
- *Repayment of family debt*

The preceding discussion traces the various stages of the migration framework and the multitude of possible traumatic stressors to which unaccompanied children are exposed. The following sections of this article will address the potential biological impacts of

adversity and trauma, and will discuss specific considerations for working with children exposed to trauma.

### 3. Trauma and Toxic Stress

The migration journeys of immigrant children from Mexico and Central America seeking protection and safety are clearly rife with experiences of significant and severe stress, adversity, and trauma. Providers working with immigrant children fleeing or experiencing danger have an understanding for what constitutes a traumatic experience (Box 6) and "toxic stress," or how these experiences "get under the skin" to influence behavior and ultimately to cause physical and mental disease and dysfunction.

It is important to understand that a traumatic experience is subjective. As defined by the U.S. Substance Abuse and Mental Health Services Administration (SAMHSA), "individual trauma results from an event, series of events, or set of circumstances that is experienced by an individual as physically or emotionally harmful or life threatening and that has lasting adverse effects on the individual's functioning and mental, physical, social, emotional, or spiritual well-being" (Substance Abuse and Mental Health Services Administration 2014, p. 7). Individual responses to trauma exposure are shaped by their developmental stage (including intellectual capacity and cognitive flexibility), as well as the availability of resources—internal or external to the individual—to help them understand and manage the situation or threat. Reduced sense of control during a potentially traumatic event corresponds with increased trauma severity and poorer psychological and health outcomes (Substance Abuse and Mental Health Services Administration 2014; Ford et al. 2009). While children who migrate, especially those who are unaccompanied, demonstrate extraordinary resilience, they may also be subject to loss of control at each stage of the migration pathway. All professionals working with immigrant children should be cognizant of this paradox and use a strengths-based approach—allowing the child to find their own locus of control in the process. Failure to do so may result in behavior that is self-defeating and can negatively impact legal and health outcomes.

**Box 6.** Definition of trauma.

| Definition of trauma (American Psychiatric Association 2013) |
| --- |
| • Direct or indirect exposure to intense and overwhelming experiences that involve threat or harm to a person's physical and/or emotional integrity |
| • Overwhelms the person's coping resources |
| • Often leads to coping mechanisms that help survive/adapt in the short run but may cause serious harm in the long run |

While "stress" or change is an integral part of everyone's life, the American Academy of Pediatrics (AAP) describes three distinct types of stress that impact the health of children and youth: positive; tolerable; and toxic (Shonkoff and Garner 2012). The AAP teaches that a child's physiologic and mental ability to adapt and learn from stress exposure depends on the ability of a caretaker to mitigate a child's worries and decrease the physiologic stress response. Toxic stress is the excessive or prolonged activation of the physiologic stress response systems in the absence of, or insufficiency of, protective relationships that reinforce healthy adaptations to stress (Shonkoff and Garner 2012). Exposure to multiple stressors or traumas over time accumulates to create a heavy psychological and physiological "load" (Shonkoff and Garner 2012; McEwen 2005; Lupien et al. 2009), particularly relevant for immigrant children seeking protection who face trauma and adversity across multiple stages of their journey. While providers should not assume that all immigrant children suffer from toxic stress, they should recognize the heightened risk associated with the migration paradigm. This is especially true of unaccompanied children or those separated at the border from their parent(s) as they do not benefit from the buffering effect from a loving caregiver.

Scientific advancements in understanding the nexus of brain development, and the impact of toxic stress adds rationale and urgency to the practice of a trauma-informed approach with immigrant children. Ongoing neurobiology research illustrates that the brain's architecture continues to change and develop after birth and into young adulthood (Blakemore and Choudhury 2006; Dahl et al. 2018; Steinberg 2009, 2014), and is modulated by neuroendocrine hormones in response to physical and perceived stimuli. Repeated and long-lasting stress or "toxic stress" can cause dysregulation of hormone levels to the extent that the neuronal architecture of the brain changes, affecting emotional processing, learning, memory and executive function and the ability of the system to self-regulate (Shonkoff and Garner 2012; Lupien et al. 2009; National Scientific Council on the Developing Child [2005] 2014; Romeo 2017; Teicher et al. 2016). In particular, brain changes associated with childhood trauma exposure result in a hyperactive or over-sensitized alarm response, leaving individuals susceptible to the emotional, behavioral, and cognitive reactions consistent with the evolutionary survival response (e.g., fight, flight, or freeze) following the mere perception or reminder (based on internal or external cues) of potential threat or danger. Prolonged stress can have "potentially permanent effects on a range of important functions, such as regulating stress physiology, learning new skills, and developing the capacity to make healthy adaptations to future adversity" (Shonkoff and Garner 2012, p. e237). Increase in other hormones can cause increased inflammatory and immunologic disease in children who are exposed to excessive stress (Chen et al. 2003; Chen and Miller 2007; Suglia et al. 2009, 2010). These cumulative effects thus have the potential to lead to physical harm and long-term health and behavioral consequences (Shonkoff and Garner 2012; McEwen 2005; Lupien et al. 2009).

The study of Adverse Childhood Experiences ("ACEs") was the first study to link the cumulative effects of toxic stressors that occur in childhood with physical health outcomes at a population level (Felitti et al. 1998). The ACEs studies delineated multiple categories of adversity that children may experience: abuse, physical and emotional neglect, and household dysfunction (Bucci et al. 2016; Chen and Miller 2007; Felitti and Anda 2010). ACEs research has found a dose–response relationship between ACEs and health outcomes such that exposure to greater numbers of ACEs in early childhood corresponds with increased severity and risk for the physiologic disruptions that cause both short-term and long-term adverse effects on children's future physical and mental health and well-being (Bucci et al. 2016; Felitti and Anda 2010). While the ACEs studies did not specifically address migration-related adversity and trauma, the ACEs categories have significant overlap with adversities experienced by immigrant children seeking protection in the US. For example, the original and subsequent ACEs studies address parental separation, physical and emotional neglect, and experiences of abuse, each of which have been identified and experienced by immigrant children across the various stages of migration.

The developmental shifts associated with ACEs and trauma exposure may result in outcomes that shape a child's cognitive, emotional, physiological, and behavioral responses to their environment (Table 1). In many cases, toxic stress exposure results in psychological and behavioral profiles that emphasize (1) immediate survival responses over long-term planning and skill development, (2) constant attentional vigilance over focused attentional control, and (3) behavioral impulsivity over behavioral regulation (Shonkoff 2016). While these responses are effective and functional adaptations in the context of a dangerous and threatening environment, they prove maladaptive or problematic in the more mundane context of a child's day-to-day school and social life (Teicher et al. 2016; Bucci et al. 2016; Shonkoff 2016; Teicher and Samson 2016). Furthermore, trauma-oriented response styles become engrained over time through chronic or repeated exposure to threat, trauma, and/or trauma reminders, as determined by principles of learning and conditioning (Hebb 1949).

**Table 1.** Adverse brain development from toxic stress can result in difficulties in multiple dimensions.

| Bodily Functions | Behavior | Development and Learning |
|---|---|---|
| • Sleeping problems<br>• Eating problems<br>• Toileting problems (very young and school age) | • Attachment: difficulty trusting, connecting<br>• Self regulation: avoidance, numbing/ dissociation, somatization, hyperarousal<br>• Affect: flat affect, incongruent with content, lack of emotional connection, lack of vulnerability, dissociation, hyperarousal<br>• Mood: anxiety, irritability, sadness<br>• Behavior, Impulse Control, and Communication: interrupts, yells easily, hits, escapes, aggression, recklessness, withdrawal/avoidance, frequent and severe tantrums | • Cognition: intrusive thoughts/ flashbacks, negative bias affecting mood, concentration difficulty, hypervigilance/distractibility<br>• Memory: Limited working memory, poor explicit recall for overwhelming events<br>• Executive Functioning Problems: difficulty with organization and problem-solving<br>• Perception of self and the world: pattern of negative thoughts, negative worldview, hopelessness, sense of helplessness, low self esteem |

Lastly, it is important to realize that the developmental stage of the child greatly influences the expression of trauma and toxic stress. This is especially significant in adolescents who might appear adult-like but think they are unique in their experiences, be embarrassed by reminders of trauma, practice risky behaviors, have poor school or work performance, and express anger and shame at themselves.[2]

### 4. The Effects of Traumatic Stress on Immigrant Children's Presentations and Functioning in Legal Interviews

Children's experiences of toxic stress can clearly affect how they present in their interactions and engagement with legal professionals, particularly as these interactions relate to the gathering of information about experiences of adversity and trauma (Box 7). Bringing a child into a legal interview may seem innocuous to an advocate or interviewer but may lead to heightened anxiety and stress in that child. Taking account of the background knowledge about trauma and toxic stress (presented above), legal professionals can be prepared to anticipate what challenges may emerge as children seeking protection engage in the interview process. While we subsequently describe some of the common presentation styles observed in immigrant children seeking protection (and other children exposed to trauma), this is not an exhaustive description. Every child is different, and traumatic stress reactions are influenced by various multi-faceted factors. Therefore, while it is important for legal professionals to be prepared around challenges expected to arise in interviewing and information-gathering, they should also expect the unexpected.

First and foremost, children with histories of trauma (especially when interpersonal in nature or when experienced within caregiving relationships) may experience difficulty in establishing trust and rapport with interviewers. This hesitance to make a trusting investment with the interviewer can be heightened when the interviewer does not have shared identity (e.g., due to gender, cultural, sexual orientation, or other differences) with the child, or when the interviewer shares identity characteristics with someone who has perpetrated harm to the child (e.g., a government official). The lack of trust in the interviewer can manifest as withdrawal, avoidance, noncompliance, or defiance in response to requests or direction from the interviewer. Such reactions should not be construed as obstructionist or ungrateful, but should be understood as a protective reaction to prior experiences of harm, abuse, and violations of trust by others.

---

[2] For further information on differential reactions to traumatic stress based on age and developmental stage, readers are referred to resources available through the Substance Use and Mental Health Services Administrations (https://www.samhsa.gov/child-trauma; accessed on 30 December 2022) and the National Child Traumatic Stress Network (https://www.nctsn.org; accessed on 30 December 2022).

The impact of trauma on cognition can lead to difficulties with attention, concentration, and memory that frequently surface during the interview process. Children may be easily distracted, tangential in focus, or challenged to recall specific memories and events. They may have a hypervigilant attentional style, perhaps demonstrating a propensity towards anxiety as they remain on the lookout for danger or risk (both due to their physical environment, and to the potential repercussions of information-sharing). These challenges may be part of a general presentation that stems from histories of trauma and adversity exposure, or they may be specifically elicited by interview content focused on experiences of stress and trauma. Dissociative responses in particular can be triggered by trauma-related content that is overwhelming.

These cognitive difficulties can impact children's abilities to provide the narrative that can be critical for the legal claim. There may be little or no elaboration on details of events or experiences, which can stem from both active avoidance of this content (due to the overwhelming nature of the material) as well as difficulties with recall. The quality of the narrative can also be impacted, as presented trauma narratives are often initially disorganized, non-linear, incoherent, or perseverative. It is not uncommon for children's behaviors and mood to appear incongruent with the quality and content of the information being discussed.

Child trauma exposure impacts children's skills with emotion identification, emotion expression, and emotion regulation. Therefore, in interviews, immigrant children may have particular difficulty in articulating their (current or past) emotional states, often times due to a lack of awareness or differentiation between different internal emotional states as well as lack of language or experience in naming their feelings. They may also be easily triggered into intense emotional reactions (generally encompassed by fear, anger, or sadness), that can escalate quickly and subsequently impact their behavior and presentation in the interview context. Such reactions may manifest in behavioral indicators of distress, such as restlessness, fidgeting, withdrawal (e.g., fleeting or absent eye contact), physiological agitation (e.g., rapid breathing), and changes in speech patterns (rapid, slowed, or pressured).

**Box 7.** Common challenges presented in interviews with children exposed to trauma.

- Difficulty with attachment—negotiating and developing trusting relationships
- Difficulty with attention, concentration, and memory
- Challenges in providing narrative
  - Little or no elaboration—avoidance and/or recall difficulty
  - Disorganized, non-linear, incoherent, perseverative
- Difficulties with emotion identification, expression, and regulation
  - Behavior and affect that are incongruent with events described
  - Behavioral indicators of distress—fidgety, restless, no eye contact, rapid breathing, fast talk, tangential

## 5. Conducting Trauma-Informed Interviews with Children in Migration

*5.1. Guiding Principles for Trauma-Informed Interviewing*

Practice approaches for conducting trauma-informed and trauma-sensitive interviews with children in migration are framed by the guiding principles and approaches for trauma-informed care systems (Substance Abuse and Mental Health Services Administration 2014). A trauma-informed system is defined as one which "*realizes* the widespread impact of trauma and understands potential paths for recovery; *recognizes* the signs and symptoms of trauma in clients, families, staff, and others involved with the system; and *responds* by fully integrating knowledge about trauma into policies, procedures, and practices, and seeks to actively *resist re-traumatization*" (Substance Abuse and Mental Health Services Administration 2014, p. 9). Furthermore, SAMHSA delineates six key principles of a trauma-informed approach: (1) safety, (2) trustworthiness and transparency, (3) peer support, (4) collaboration and mutuality, (5) empowerment, voice, and choice, and (6) attention

to cultural, historical, and gender issues (Substance Abuse and Mental Health Services Administration 2014, pp. 10–11).

Adherence to the principles of trauma-informed systems means taking active efforts and maximizing opportunities within the context (and constraints) of the interview to:

1. ensure that children feel physically and psychologically safe during the interview, and that any perceived risks are addressed (*safety*);
2. develop a clearly defined and trustworthy relationship and rapport, where interviewers provide clear and accurate information to children (*trust and transparency*);
3. develop plans and efforts for obtaining support from peers, family, and community (*peer support*);
4. align efforts and priorities so that the child and interviewer are working together towards a shared goal (*collaboration and mutuality*);
5. provide children with maximal control over the interview process and highlight children's (and their family's) strength and resilience (*empowerment, voice, and choice*); and,
6. acknowledge and integrate information about children's developmental stage, trauma history, and cultural identity into the interview process (*attention to cultural, historical, and gender issues*).

By adhering to these principles and aspirational guidelines, and by further integrating a child-sensitive developmentally tailored approach, providers can uphold children's best interests in the context of advocacy and humanitarian work, in accordance with the protection of children's rights (United Nations 1989). The practice guidelines below are drawn from established best practice in conducting forensic interviewing of children, as applied to the context of children in migration (Lyon 2014; Quas and Lyon 2019; National Institute of Child Health and Human Development 2014).

*5.2. Interview Practice Recommendations*

Our practice recommendations and guidelines are intended to set aspirational standards and goals that are pursued as part of a continuous process and investment for interviewers, both within their engagements with specific children and over the course of their professional journey. We recognize that (1) many interview contexts constrain opportunities to achieve these aims, (2) there are no guarantees or certainty of outcomes in this work, and (3) as professionals, we are never done learning or building skills. Many interview processes with immigrant children are constrained by resource limitations, insufficient time, and interview contexts (e.g., children who are currently detained, children in the process of migration facing ongoing risk). Therefore, interviewers must accept the presence of the uncertainties and make ethical determinations of risk/benefit tradeoffs within the conditions and constraints of each interview, and engage in continuous efforts to build skills and knowledge as a professional responsibility. The acceptance of uncertainty also requires understanding and acknowledgement that there is no "one size fits all" solution for all children in all interviews. Trauma-informed interviewing requires that we remain flexible in our approach in order to effectively and sensitively respond to the diversity of child presentations and contexts. For example, while open-ended questioning is one strategy frequently recommended for avoiding leading questions or introducing bias, some children require close-ended questions (with response options) based on their developmental level and need for concrete information. Pediatric and mental health professionals have specific experience and training navigating these challenges and thus should be consulted, included, or engaged in legal interview and information-gathering processes, as discussed in subsequent sections of this article.

5.2.1. Obtain Background Knowledge

In order to conduct a trauma-sensitive interview and to adequately understand children's context, interviewers must often take efforts to conduct research, obtain training, and gather background knowledge related to children's identity and history. The approach

to interviewing children must take into account their age, developmental stage, level of education, cultural background and history of trauma. Practitioners should make an investment to obtain general knowledge about foundational principles, science, and facts relevant to all children in migration (e.g., knowledge about child development, information on the impact of child adversity and trauma). There is also background research that is often necessary for informing engagement with individual children relevant to their specific case (such as knowledge about their identified culture, community history, or impacts of specific types of traumas experienced), which especially includes the migration paradigm relevant to their community. Much of this case-specific work can be done in preparation prior to an interview, but also commonly takes place during the course of the interview process, as new information about history and identity surfaces. This effort is rooted in practitioner humility in which their approach is guided by an awareness that "I can't know all there is to know about this child's experience." More specifically, the domains of information-gathering that are particularly relevant for immigrant children include:

- child developmental processes and the impact of development on children's cognitive, emotional, and interpersonal functioning;
- impact of compounded trauma and adversity exposure on children and child development;
- information about children's primary language, culture(s) and community of origin, including the history of their communities;
- level of education and literacy (within the community of origin and specific to the child);
- migration experience and trajectory (e.g., accompanied, unaccompanied, separated);
- current legal status and related risks;
- current living placement/situation and related risks (given that children's presentation and level of disclosure may be impacted by how safe they feel in their current situation).

Gathering background information in these domains helps the interviewer to know what to anticipate in the interview, understand the diverse presentations and behaviors that may arise, respond in an effective and supportive fashion, and identify and capitalize upon children's strengths and resilience. This background information may also inform additional resources that need to be brought into the interview process, such as interpretation or cultural brokerage supports. These process improvements in turn can improve the outcomes of the interview, as well as the longer-term legal, health, and functional outcomes for the child.

### 5.2.2. Build Rapport and Engagement

The success of the interview is determined by the quality of the working relationship between the children and the interviewer (Lyon 2014). Therefore, it is imperative to invest in relationship development and rapport-building. While many interview contexts can limit opportunities for this relationship investment, it is nonetheless critical that an investment is made in this effort, and that whatever opportunities present are maximized. This effort is especially critical given that children are particularly sensitive and responsive to the adults around them, and that children who have endured trauma often have difficulty establishing trust and safety with others (as an otherwise adaptive response to violations of trust and safety within relationships). A general underlying principle is that the interviewer recognizes and engages with the child *as the full individual that they are*; that is, for example, they are more than a victim of trauma, they are more than an individual from "the Northern Triangle," they are more than an asylum-seeker, and they are more than an interview subject. Ways to accomplish this relationship development and rapport-building, include:

- using a friendly, genuine tone and avoid an overly professional or rigid approach;
- using humor, warmth, and personal connection;
- showing interest in children's interests and experiences (asking about their day, their current well-being, their hobbies, their family, etc.);

- building engagement around non-threatening (i.e., not trauma-related) topics with easy-to-answer questions;
- providing multiple opportunities for meeting and connecting to gradually build the relationship and demonstrate consistent presence over time;
- identifying and connecting around commonalities, interests, experiences shared between the child and interviewer (as appropriate—always consider the risk/benefit tradeoffs of personal disclosures);
- considering and addressing power imbalances in the child-interviewer dynamic (e.g., due to ethnicity, gender, physical stature, disability; for example, seating at equal heights so not to intimidate children);
- continuously working to explore and identify child priorities and core goals and values (e.g., what are they most concerned about/what do they need most *in this moment*?).

5.2.3. Maintain Respect for Children, Their Family, and Community

Each of the practice recommendations described throughout must be driven by an utmost respect for the child, their family, and their community. Operating from a perspective of respect and acceptance helps to facilitate the process of trust and relationship development, and also motivates relevant information-gathering that affects legal processes and outcomes. As mentioned previously, practicing humility—in particular cultural humility—is a foundational element of respecting the child and their background, and is critical for understanding their priorities and actions. *Cultural humility* holds that "the client is uniquely qualified to educate the [interviewer] about his or her multiculturalism, that is his or her membership in multiple cultural groups and life stressors, which in turn affects [treatment, interview, and advocacy] priorities" (Falicov 2014, p. 31).[3] Further, this practice entails balancing the risks between "underestimating the impact of culture and incorrectly attributing dysfunction to a pattern that is normative in the individual's or family's culture" and "overestimating and magnifying the importance of culture at the expense of failing to recognize dysfunctional individual or family processes" (Falicov 2014, p. 31). At its core, cultural humility requires that we approach each interview and interaction with curiosity as an opportunity for learning and connection, and that we avoid making assumptions about an individual based on their cultural background, and vice versa.

Maintaining respect for the child also involves adopting a *client-centered approach* throughout the interview process and interactions. This means adopting a stance of *unconditional positive regard* (Bozarth 2013; Quas and Lyon 2019) and *radical acceptance* (Linehan 1993; Brach 2003) for whatever presentations, challenges, ruptures, or obstacles may present in the course of the interview relationship. Children with trauma histories can often present with seemingly odd and puzzling behaviors that can interrupt and interfere with interview goals; it is important to remember and accept that these challenges are often functional adaptations to severe threat and malevolent circumstances, and that the resulting presentations and behaviors serve a purpose and function from the perspective of the child's experience. Furthermore, in the absence of cultural humility and unconditional positive regard, culture-based differences between children and interviewers in values priorities and interactional style can emerge as challenges or points of conflict. Adopting unconditional positive regard also helps to identify and elicit client strengths, as challenging trauma-related behaviors and presentations are reframed as qualities that previously helped children survive adversity.

The client-centered approach treats *children as experts in their own experience.* They are best-suited to find, identify, and advance the most effective ways to tell their story. This means that interviewers must be flexible in exploring a variety of modalities for information-gathering, which may include the use of art, play, dance, references to popular culture, and other forms of creative expression throughout the interview process; a rigid

---

3 Here we are broadening Falicov's definition of cultural humility within therapeutic contexts, to be applied within general service provision, including interviewing for legal/advocacy purposes.

reliance of direct interviewing and narrative story-telling can be counterproductive to the interview process, and, in many cases, is culturally incongruent and developmentally inappropriate. Furthermore, children must be allowed to tell their stories in their own words and share their own subjective perspective without judgment or correction; while the ultimate goal of the interview process may be to elicit a coherent and linear narrative that reflects an "objective truth", this is an outcome that is reached gradually wherein the child's report at a given moment is viewed as one piece of information that informs a broader objective narrative.

The client-centered interview approach is rife with *reflection and validation* of the child's experience throughout. Reflective communication is critical, such as use of active and empathic listening skills that utilize both verbal and nonverbal communication strategies. This may include:

- body language that demonstrates attunement with the child's mood and experience (e.g., relaxed and open posture, gradually increasing physical proximity as a demonstration of support and interest, appropriately mirroring a child's affect and presentation);
- open reflection and identification of children's observed mood and emotions;
- offering summarizing statements that reflect content and affect;
- checking for understanding and confirmation of children's statements and feelings, also offering opportunities for children to provide correction, clarification, and elaboration;
- inviting children to pose their own questions;
- following and mirroring children's pacing and tone (more often than not, this means slowing down);
- validating children's experiences of reported and/or observed emotional distress or cognitive challenge (e.g., attention or memory difficulty) as common, normal, and often protective responses to trauma and adversity;
- responding sensitively and appropriately to children's expression of distress (while interviewers also regulate their own secondary affect);
- allowing or creating time and space for silence and reflection;
- acknowledging and reflecting (instead of avoiding or minimizing) experiences of feeling stuck or hopeless (even if it requires the interviewer to acknowledge that they themselves "don't have the answers").

Efforts to provide reflection and validation throughout interviews help children have an experience in which they feel seen, heard, and understood, which is a critical element for effective interviewing, and also for their general well-being. Indeed, when done well, trauma-informed interviewing of children can be a restorative and therapeutic experience that helps children progress in their paths towards healing and recovery. We must acknowledge that, in many cases, immigrant children are telling their stories (and trauma histories) for the first time and in new ways, meaning that we have a critical opportunity to provide a corrective and validating experience which empowers children around content and histories in which they have previously been disempowered or marginalized. Aligned with the client-centered approach, children's sharing of sensitive histories and emotional material warrants an expression of gratitude for their courage, effort, and emotional burden in this process.

### 5.2.4. Maximize Agency and Predictability

At its core, trauma and traumatic stress results from having limited information and perceived agency while facing adversity and potential threat. The subjective nature of trauma means that things become problematic when the impacted individual has the experience that they do not know what is happening (to themselves or to a beloved individual or community), and they feel that they have no control over themselves and/or the outcome. In this way, interviews about trauma content (which elicit trauma-related cognitions and affect) can either be re-traumatizing and disempowering, or they can be

restorative and empowering, largely dependent on whether the interviewee feels a sense of predictability and agency through the process.

One primary means of maximizing agency and predictability is through efforts to *provide clear information and expectations about the interview process* (Lyon 2014). At the outset, children should be oriented to the parameters of the interview, including length, process, and content. They should be prepared for the interview content areas, once again acknowledging and normalizing that they may experience emotional reactions or distress when discussing experiences of adversity and trauma; they should be provided with the opportunity to anticipate, prepare, and plan for potential emotional reactions that may surface (usually centered around fear, anger, or sadness). Children should also be helped to understand expectations for the nature of the working relationship. What support can be offered within the scope of the interviewer's capacity? What will be the duration and nature of the relationship, and the expectations for contact? For children, the provision of clear information often requires that interview parameters are presented in developmentally appropriate language and that interviewers check for understanding and consent. Ultimately, this means that our aim as interviewers is to avoid any surprises (in terms of process, content, and/or children's reactions) for children during the interview process; if children are caught off-guard or become unexpectedly dysregulated during the course of the interview, this is an indication that we (as the interviewer) missed an opportunity to provide clear information or predictability.

Maximizing agency also means that interview goals *align with children's goals, values, and priorities*. Describing interview parameters also involves providing clear rationale for the interview—how will the interview outcomes align with children's goals for themselves, their family, or their community? This alignment often needs to be made explicit and demonstrated clearly for children in order to enhance their motivation and engagement in the interview process. Additionally, when there are ruptures, breaks, or challenges in the interview process, re-visiting the shared goals and aligned priorities often serves as the foundation to return to the collaborative work of the interview. With this, we must recognize that children's goals and priorities may shift over the course of the interview due to changes in insight, context, status, outcome, or other life and personal circumstance. Therefore, goal alignment is a topic worth revisiting throughout the process.

Efforts to maximize agency and predictability require interviewers to also *maximize child decision-making and control over the interview process*. Children should be given every opportunity possible to set the interview context and environment; this may include determining who is present for the interview, who sits where, and when and how the interview takes place. Providing for children's basic needs also helps to increase sense of agency and control, such as through offering food and drinks, providing easy access to bathrooms, and eliminating safety risks. Similarly, children should be provided access to child-friendly resources for coping and creative expression, such as art materials, fidget toys, stress balls, and relaxation activities. Children should be allowed to set the pacing of the interview, including being informed at the outset and also being frequently reminded that they can ask to pause, take a break, or stop the interview at any point (each reminder to this end can help to increase the child's sense of control and empowerment). Interviewers are strongly recommended to check in regularly with children about the interview process and their present-moment emotions and well-being. These pauses to check in and reflect allow children to temporarily disengage from stressful content, to engage their executive functioning and self-regulation capacities, to have an opportunity to express themselves, and to have an opportunity to seek support or request resources. This critical process step is easier said than done, as interviewers are often feeling pressed for time and/or constrained by contextual factors; nonetheless, the benefits of the "regular check-in" are tremendous.

Additional considerations involve creating an environment that feels safe and is child-friendly. A space that invites comfort and play, with relaxing imagery and availability of toys, crayons, and other objects for creative expression and coping, can improve the process, in comparison to the commonly cold, stark environments of a traditional conference room

or office. The set-up should ideally ensure a sense of freedom and safety, by providing an open space (perhaps even leaving the door open, barring confidentiality concerns) or positioning children so they are facing the door. At the same time, it is important to ensure a sense of confidentiality and privacy, and giving children the opportunity to be interviewed alone to allow for sensitive information to be disclosed (e.g., abuse, LGBTQ identity).

An especially important opportunity for maximizing agency and predictability for immigrant children relates to the processes of presenting at an administrative interview or immigration court hearing. While many of the circumstances and conditions of such interactions may be out the child's (or attorney's) control, it is nonetheless important to help them anticipate and prepare for the interview or hearing. Careful, developmentally appropriate explanation of what to expect and how to manage interview or testimony experiences can serve to increase children's sense of agency and predictability. Particularly since the material is focused on trauma experiences, children should be prepared (as much as possible) for what they will be presenting and the related emotional reactions they may experience. Rehearsal and elaboration of immigration/trauma narratives can be helpful to this end. Attorneys and legal professionals should try to apply the principles and practices discussed throughout this section in the administrative or court setting, and should be prepared to advocate for accommodations in typical procedures based on the needs of their specific clients.

Finally, maximizing agency and predictability also require providing *access to resources and supports*, both in the moment of the interview, and as follow-up. Safety concerns should be addressed promptly, supportive resources should be offered or pursued (making attempts to advocate for resources even in the context of multiple constraints), and additional referrals (e.g., to other providers or resources) should be made, as appropriate. It is not necessarily within the capacity or role of the interviewer to engage in the provision of service delivery (especially for services outside of their professional scope), but every interviewer can make an attempt at connecting the client to resources, which helps increase the child's sense of stability and control in their current context.

### 5.2.5. Provide Closure to the Interview and Engagement Process

In most cases, legal and advocacy interviews for children in migration are a time-limited process, and it is important to provide closure as a means of containing this process. This includes closure around completion of a specific interview (or series of interviews) as well as closure to the working relationship. Closure processes involve summarizing (sometimes briefly) both the information obtained through the interview, and the utility and potential impact of this content. It is helpful to remind children of how their engagement and effort may benefit themselves or their community, in alignment with their individual goals and the interview rationale. In this effort, interviewers can aim to instill hope for a good outcome and better future, but it is important not to overpromise or raise expectations for an outcome that cannot be guaranteed. It is also highly beneficial to comment on observations of the child through the process, in particular highlighting the strengths, positive qualities, and resilience factors that surfaced during the course of the interview and working relationship. This helps to return to the connection with the individual (beyond their trauma) and also to help them feel empowered by connecting with the qualities that have helped them survive adversity. Sometimes it can be helpful to share something that the interviewer will take from their relationship with the child (and perhaps bring to other children in the future). Closure processes should also include efforts to "return to baseline" after addressing challenging content; this could include opportunities for play, recreation, relaxation, celebration, or personal connection. These "return to baseline" activities help to contain the challenging content and emotions, though children (and their caregivers) should also be equipped with the knowledge that the challenging material may continue to resurface in the hours, days, and weeks following the interview as part of a natural response to trauma. Children and their caregivers should prepare themselves— and be prepared—accordingly with resources and follow-up plans and supports, should

this distress prove problematic. Similarly, closure involves a review of next steps, which could include potential connection to additional referrals or other resources in their current context. The "next steps review" should also summarize what will happen next with the interview material, including the legal or advocacy process, once again reminding children what will come of their efforts and engagement. Finally, ending on a note of gratitude and appreciation is recommended. Once again, children have put effort into a collaborative process that not only advances their personal and family goals, but also helps the interviewer do their job and achieve their goals. The recognition of this effort and contribution should not go understated: the trauma-focused interview is hard work that does not come without emotional labor for the child interviewee, therefore, interviewers have a responsibility to honor the efforts, contributions, and sacrifices of children and their families with gratitude, appreciation, and respect.

5.2.6. Maintain Professional Balance and Engage in Self-Care

An overarching need and element of a trauma-informed interview is a stable, compassionate, and regulated interviewer. Individuals and professionals engaging with clients exposed to trauma are at risk for secondary traumatic stress, vicarious traumatization, and burnout (National Child Traumatic Stress Network 2011). Therefore, it is important that professionals maintain an appropriate balance in their professional activities so that they are not overwhelmed by the trauma content and related emotional responses inherently involved in trauma-focused interviewing. This self-care is more than a privilege: it is a professional responsibility that is necessary for sustaining oneself in this work, and for providing a supportive, well-regulated experience for interviewees that minimizes risk for re-traumatization. Preventing the negative impacts of secondary traumatic stress involves: (1) awareness of personal secondary stress reactions and burnout indicators, (2) engagement in self-care practice and appropriate professional balance, and (3) engagement in support and consultation from peers and mentors. Secondary stress reactions are an inevitable experience in working with children exposed to trauma, as this is a natural consequence of being a caring individual who maintains presence and connection with others. By maintaining professional balance and engaging in self-care, we give ourselves the opportunity to experience the vicarious resilience (Hernandez et al. 2007) that comes from working with survivors of trauma.

**6. Applying Trauma-Informed Practice in a Medical—Mental Health—Legal Partnership Approach**

Immigration attorneys like all professionals that work with newly arrived immigrant children—unaccompanied or in families—must take an approach that keeps the best interests of the child in mind. Minimizing the re-traumatization of children while also getting important details of their history is a delicate balance. By recognizing the signs and symptoms of trauma, legal professionals can improve the care their clients receive by referring them to medical and mental health experts and social services. Additionally, working collaboratively can enhance the physical, emotional and legal outcomes (Figure 3).

Medical–mental health–legal partnerships "integrate the unique expertise of lawyers into health care settings to help clinicians, case managers, and social workers address structural problems at the root of so many health inequities." (National Center for Medical Legal Partnerships 2022). These partnerships have been found to enhance health outcomes, decrease stress and improve mental health, have more stable housing and improved finances. Moreover, medical and mental health evaluations have shown to improve legal outcomes in adult asylum cases (Lustig et al. 2008; Atkinson et al. 2021). In one study, 89% of adult asylum seekers with a medical evaluation won their case compared to only 38% of those who did not have an evaluation (Lustig et al. 2008).

An important aspect of this practice is to minimize harm through re-traumatization which unwittingly may occur through multiple interviews. Children may become distressed in the process of being questioned multiple times, by multiple individuals (often

strangers) from different backgrounds and disciplines about events in their past. Referral and coordination with pediatric and child mental health professionals can ameliorate these risks. Instead of trying to repeatedly get a history of the events that led the child to escape their country, referring a child to a mental health or medical expert may help minimize re-traumatization and allow for histories of trauma to unfold; importantly, in a controlled trauma-informed setting. Mental health and pediatric specialists have the professional training and tools needed to provide a safe environment and interventions for children who may be triggered and decompensate when recounting their histories. Further, detailed expert testimony and written evaluations by child protection experts (pediatricians, social workers, psychologists and psychiatrists) can corroborate histories of trauma, provide evidence (e.g., X-rays, photographs and explanation of scars, mental health assessment scores) and prevent re-traumatization due to repeated testimony by the child. These are principles that have been used in child abuse and neglect cases (Pantell 2017).

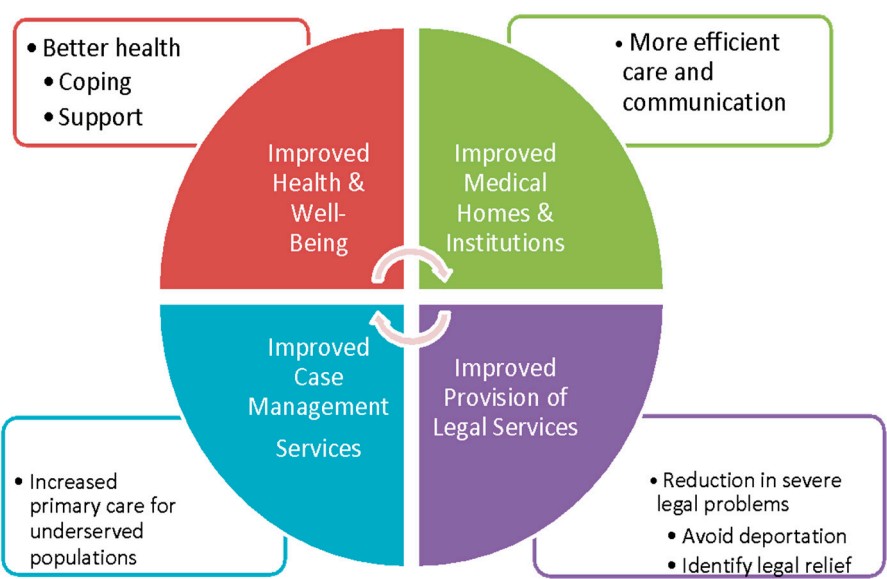

**Figure 3.** Outcomes of effective medical–legal partnerships.

Teo's case (Box 8) illustrates many of the essential points made above. Teo's lawyer sensed her client's stress, knew her own limitations and also realized he was avoiding her. Her agency was part of a medical–legal partnership program and was able to get Teo a timely appointment. The referral successfully resulted in improvement in her client's psychological and physical well-being, as well as obtaining the information needed to win his immigration case.

In summary, medical and mental health specialists have a unique role in children's immigration court proceedings and can provide trauma-informed care that can both lead to successful health and legal outcomes. Key points include:

- Important role of the medical home in which children and their caretakers build a trusting long-term relationship.
- A trauma-sensitive setting that understands the developmental and psychological needs of children becomes a safer setting in which to reveal complex histories of trauma.
- Understand that children may not be comfortable in a law office setting and that may interfere with a child's sense of safety. Legal teams should develop strong networks or partnerships with medical and mental health experts to facilitate timely referrals.
- Legal provider should try to identify healthcare facilities that have integrated co-located mental health services to minimize need for client to go to different offices and to allow for a warm "handoff" that can improve adherence to mental health services.

- Collaborating with medical and mental health experts can mitigate or prevent re-traumatization by having to repeat traumatic histories repeatedly to different professionals.
- Medical–mental health–legal partnerships can help prepare youth for court and mitigate the need to repeat their histories to judges by providing written and oral professional testimony.

**Box 8.** Case example.

Teo is an adolescent male from a K'iche Indigenous community in Guatemala who fled the country for his safety. Teo initially represented himself at the USCIS asylum office interview but was referred to immigration court because he provided insufficient information to demonstrate that he qualified for asylum. He told his new attorney for his immigration court case that his village was no longer safe for him to stay there but declined repeatedly to provide any further details about the reasons for this statement. Instead, he complained at multiple visits about insomnia, headaches and tooth pain. His attorney was concerned that there was more to Teo's history and repeatedly would ask him to tell more of his history to her, but he would not and eventually began missing appointments and avoiding her phone calls. Because he was suffering from a toothache and insomnia, he agreed to make an appointment at a healthcare center, that also specialized in working with unaccompanied minors. He was slowly building rapport with his pediatrician, but it was not until the pediatrician accompanied him to a specialty appointment, that a deeper trusting relationship was formed. He then began to disclose that his insomnia stemmed from recurrent nightmares. Upon further questioning, he disclosed the cause of those nightmares. He recounted that at 15 years of age, he and his best friend were accosted by masked criminals on their way home from school. Teo remained passive, but his friend fought back and was brutally murdered in front of him. He was allowed to go free but was told that he would be killed if anybody found out what happened. Fearing for his life, he kept this secret to himself and never told another soul until that pediatric visit. His pediatrician referred Teo to the program's psychologist and over the course of 9 months of intensive treatment, agreed to disclose the traumatic event to his lawyer. Eventually, his psychologist and lawyer worked together to prepare him for court. Teo was able to successfully recount his history to a judge and along with oral and written testimony by the healthcare team, won his case and was granted asylum.

## 7. Conclusions

As illustrated in the current article, protecting the rights of children throughout the immigration process requires application of trauma-informed practice while conducting the necessary work of interviewing children about their migration journey and trauma experiences, which in turn requires a multi-disciplinary collaborative approach for engaging with children seeking protection. All aspects and elements of this work require core emphasis on protecting and promoting the child's best interests. Implementing these practices and recommendations can improve children's experiences during trauma-focused interviewing, and can also improve legal, health, and social outcomes both in the short- and long-term. As pediatric and mental health professionals committed to the care and long-term health of immigrant children seeking protection in the US, we have benefited greatly from our collaborative work with legal professionals, in which we have gained critical knowledge and information about children's experiences in legal processes in the immigration system, and which has helped us serve as effective advocates for the children and communities we care for. We hope that we can effectively offer guidance and input for legal professionals that stems from our training and professional disciplines, as part of the collaborative interdisciplinary exchange (for further information on topics discussed in this manuscript, please see Box 9). We firmly believe that we are stronger together, when we break out of our disciplinary systems and silos, and work collaboratively towards collective impact that strives to protect the rights and well-being of children in migration.

**Box 9.** Resources.

> Stanford Digital Medic Trauma Informed Interviewing Techniques: https://digitalmedic.stanford.edu/our-work/trauma/; accessed on 4 January 2023.
> National Center for Medical Legal Partnerships: https://medical-legalpartnership.org/; accessed on 4 January 2023.
> American Academy of Pediatrics Toolkit for Immigrant Child Health: https://www.aap.org/en/patient-care/immigrant-child-health/; accessed on 4 January 2023.
> National Child Traumatic Stress Network: https://www.nctsn.org/resources; accessed on 4 January 2023.
> Harvard Center for the Developing Child Guide to Toxic Stress: https://developingchild.harvard.edu/guide/a-guide-to-toxic-stress/; accessed on 4 January 2023.
> National Center for Youth Law Guidance for Mental Health Professionals Serving Unaccompanied Children Released from Government Custody: https://youthlaw.org/resources/guidance-mental-health-professionals-serving-unaccompanied-children-released-government; accessed on 4 January 2023.
> Centers for Disease Control and Prevention Adverse Childhood Experiences: https://www.cdc.gov/violenceprevention/aces/index.html; accessed on 4 January 2023.

**Author Contributions:** All authors contributed to the conceptualization, literature and best practices review, and writing (including original draft preparation, review, and editing) for this manuscript. All authors have read and agreed to the published version of the manuscript.

**Funding:** The authors received no external funding for the production of this manuscript.

**Institutional Review Board Statement:** Not applicable.

**Informed Consent Statement:** Not applicable. All case examples have been de-identified and anonymized, with each specific example representing an amalgamation of real-life examples and details from cases the authors have encountered in their work.

**Data Availability Statement:** Not applicable.

**Conflicts of Interest:** The authors declare no conflict of interest.

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
