# Peer review of "Pediatric Perspectives and Tools for Attorneys Representing Immigrant Children: Conducting Trauma-Informed Interviews of Children from Mexico and Central America"

_laws, 2022_

Round 1

Reviewer 1 Report

This article provides helpful background on compound trauma that may impact children from Central America and Mexico seeking asylum in the United States and how trauma-informed medical/legal collaboration may assist attorneys representing such children in their immigration court cases. The recommendations for trauma-informed interviewing are practical and provide guidance to attorneys who may be new to representing children who experience complex trauma. This is a valuable piece and contributes to the literature on the topic. 

This article mentions children testifying before a judge in immigration court proceedings. However, unaccompanied children who are in removal proceedings typically present their asylum cases in a "non-adversarial" interview before a USCIS asylum officer. It was unclear to me in some instances whether the child had already underwent an asylum interview and was presenting their case before an immigration judge, or if the scope of this article is focused on immigration court proceedings rather than the affirmative process. Since most unaccompanied children will have interviews before an asylum office, I think it would be helpful to include adjudication before USCIS. I don't think revisions need to be extensive, but include a few sentences about the process, acknowledge that the child may be before an "adjudicator", and discuss any differences between the two processes that might be helpful to note in the recommendations section. 

Regarding Box 1, Rigoberto's case - if this article focuses on children and Rigoberto entered the US on his own, then I would consider changing his age to 17, or noting that while he's now 18, he was 17 when he entered the US. I would also consider capitalizing "Indigenous". Also, it may be helpful to clarify the landowners of the coffee plantation were searching for him. 

In the final draft, double-check that the abbreviation U.S. is used as an adjective and US is used as a noun. There was some inconsistency in the article, e.g. line 22 and Box 3 case example, line 4.

Box 4 Case example, line 2 - perhaps cleared to say "his father's second wife" rather than "the father's second wife"?

Line 313 - this is one example discussing children testifying before a judge. It seemed like this paragraph may refer to children who came with their families, in which case they would be testifying before a judge if they presented at a port of entry and indicated they wanted to seek asylum, but the process of seeking a durable or permanent legal status as a stressor could also be applicable to unaccompanied children so you may want to consider broadening this paragraph to apply to both children with families who may be before a judge, and unaccompanied children who may first have an interview with an asylum officer, but if the asylum officer does not approve their case, they can get another "bite of the apple" by applying for asylum before an immigration judge.

Line 320 - perhaps consider rewording "what exactly trauma is biologically". I had to re-read this sentence a few times because I didn't initially understand what it meant. 

I think section V, Conducting Trauma-Informed Interviews with Children in Migration is very helpful both to set the principles for trauma-informed interviewing as well as provide concrete best practices. It may be worthwhile to provide some tips on a trauma-informed approach to preparing children for testifying at an asylum interview or immigration court case as that is another critical piece for attorney representing immigrant children. However, if it feels too lengthy to include an entire section on this, I would suggest at least flagging the importance of preparing children for testimony and minimizing surprises when they have to present themselves before an adjudicator as the child's reaction and comfort level may be different than at an interview with an attorney if they do not know what to expect. 

Box 8 Case Example - I would consider capitalizing "Indigenous" in this case example as well. It's unclear how old Teo was when he came to the US, but since this article is about children, as an attorney reader, I would assume he was under 18. I think it's great to provide an example with the psychologist and lawyer working together to prepare him to testify, but if he was under 18 when he arrived and an unaccompanied child, even if he was in removal proceeding, he could still proceed with his case before the asylum office first rather than court. If you want to keep this in court, perhaps the case example could be revised so that he represented himself at the asylum office interview but was referred to the court because he did not provide sufficient details to show that he qualified for asylum. Then he obtained an attorney for his immigration court case. 

Reviewer 2 Report

 Line 354 - Is there an automatic assumption to be made here that all are suffering from toxic stress?  Definition of other types of shock are not offered. 

Table 1 - are the lines in the table meant to align or are the columns giving independent info?  

415 - important issue regarding adolescent behaviour addressed and they are an often-unrepresented group.  I understand the limited word count but is there potential scope to put some references in here that shows the difference between the adolescent group and younger children/adults? 
